# Developing and externally validating a machine learning risk prediction model for 30-day mortality after stroke using national stroke registers in the UK and Sweden

Wenjuan Wang ,[1] Josline A Otieno ,[2] Marie Eriksson ,[2] Charles D Wolfe ,[1] Vasa Curcin,[1] Benjamin D Bray[1]

¹Department of Population Health Sciences, King's College London, London, UK
²Department of Statistics, Umeå University, Umea, Sweden

**Correspondence to**
Dr Wenjuan Wang;
wenjuan.wang@kcl.ac.uk

## ABSTRACT

**Objectives** We aimed to develop and externally validate a generalisable risk prediction model for 30-day stroke mortality suitable for supporting quality improvement analytics in stroke care using large nationwide stroke registers in the UK and Sweden.

**Design** Registry-based cohort study.

**Setting** Stroke registries including the Sentinel Stroke National Audit Programme (SSNAP) in England, Wales and Northern Ireland (2013–2019) and the national Swedish stroke register (Riksstroke 2015–2020).

**Participants and methods** Data from SSNAP were used for developing and temporally validating the model, and data from Riksstroke were used for external validation. Models were developed with the variables available in both registries using logistic regression (LR), LR with elastic net and interaction terms and eXtreme Gradient Boosting (XGBoost). Performances were evaluated with discrimination, calibration and decision curves.

**Outcome measures** The primary outcome was all-cause 30-day in-hospital mortality after stroke.

**Results** In total, 488 497 patients who had a stroke with 12.4% 30-day in-hospital mortality were used for developing and temporally validating the model in the UK. A total of 128 360 patients who had a stroke with 10.8% 30-day in-hospital mortality and 13.1% all mortality were used for external validation in Sweden. In the SSNAP temporal validation set, the final XGBoost model achieved the highest area under the receiver operating characteristic curve (AUC) (0.852 (95% CI 0.848 to 0.855)) and was well calibrated. The performances on the external validation in Riksstroke were as good and achieved AUC at 0.861 (95% CI 0.858 to 0.865) for in-hospital mortality. For Riksstroke, the models slightly overestimated the risk for in-hospital mortality, while they were better calibrated at the risk for all mortality.

**Conclusion** The risk prediction model was accurate and externally validated using high quality registry data. This is potentially suitable to be deployed as part of quality improvement analytics in stroke care to enable the fair comparison of stroke mortality outcomes across hospitals and health systems across countries

## STRENGTHS AND LIMITATIONS OF THIS STUDY

⇒ The models were built with robust approaches and reported according to the TRIPOD reporting guidelines with several terms adjusted for machine learning studies such as hyperparameter tuning.
⇒ Final models were published and have been made freely available for download for future external validation and use.
⇒ Death outcomes were limited to inpatient mortality and was not possible to ascertain deaths occurring outside hospital within 30 days for Sentinel Stroke National Audit Programme (SSNAP) data.
⇒ The predictors were limited to the variables that are available in both SSNAP and Riksstroke.
⇒ The models were built and externally validated with very large and comprehensive stroke datasets from a nearly complete population of hospitalised stroke in England and Wales (SSNAP) and Sweden (Riksstroke).

## INTRODUCTION

Risk prediction models are valuable tools in benchmarking healthcare outcomes across different hospitals or countries, thereby contributing to improved clinical practice.[1] These models are required to adjust for differences in the case mix of patients to allow for an unbiased comparison of stroke outcomes and can be used to identify patients at high risk of poor outcomes. Benchmarking and comparing stroke outcomes across hospitals and health systems is widely used as part of quality improvement initiatives and quality registers in stroke,[2] but depends on accurate adjustment of case mix differences to be valid.

Prediction models using machine learning (ML) algorithms are now being widely used in healthcare (e.g., in the interpretation of medical imaging data) but the potential of ML to improve the accuracy of analytics used

for measuring and comparing the quality of stroke care services is largely unexplored. Moreover, a systematic review on the application of ML methods in predicting outcomes of stroke based on structured data found that few studies met basic reporting standards for clinical prediction tools.[3] Major issues included lack of external validation, and the non-availability of the final model for external validation.[3] External validation (i.e., evaluating the accuracy of the predictions in data from a different source that used to develop the model) is especially important in assessing the generalisability of the model before it is used in other settings.

Traditionally, logistic regression (LR) has been used for risk predictions of stroke outcome. It has been suggested that ML might outperform clinical prediction models based on LR because they make fewer assumptions and can learn complex relationships between predictors and outcomes.[4] However, there are also studies showing that no evidence of superior performance of ML over LR and improvements in methodology and reporting are needed for studies that compare modelling algorithms.[5] In a previous study, we compared an updated version of LR models from Bray et al. 2014[6] and an ML algorithm (eXtreme Gradient Boosting, XGBoost) for predicting 30-day mortality after stroke and found that XGBoost performed the best.[7] Existing scores for 30-day mortality prediction, for example, PLAN[8] and iScore[9] were not compared in the previous study due to lack of certain variables.

Models developed and validated in one nation might not perform well with another nation due to the differences in the population and health systems. In this study, we aimed to develop and externally validate a generalisable risk prediction model for 30-day mortality after stroke using large nationwide stroke registers in the UK and Sweden. The intention was to develop a model developed and externally validated in high quality registry data which could be made publicly available to support stroke research (e.g., cross-national or multicentre comparisons of stroke mortality) or be deployed as part of quality improvement analytics initiatives in stroke.

## METHODS

This study is reported according to the transparent reporting of a multivariate prediction model for individual prognosis or diagnosis (TRIPOD) guidelines: transparent reporting of a multivariable prediction model for individual prognosis or diagnosis.[10]

### Data source

#### Data from Sentinel Stroke National Audit Programme for model development and internal validation

Data for developing the models were from Sentinel Stroke National Audit Programme (SSNAP), the national registry of stroke care in England, Wales and Northern Ireland. SSNAP is a Healthcare Quality Improvement Partnership (HQIP) register for stroke care quality improvement.

Data were collected prospectively, validated by clinical teams and entered into the SSNAP database via a secure web interface. SSNAP includes an estimated 95% of all adults admitted to hospital with acute stroke (ischaemic or primary intracerebral haemorrhage) in England, Wales and Northern Ireland. The original dataset from SSNAP was collected from 333 teams across England, Wales and Northern Ireland between 1 April 2013 and 30 June 2019. For the generalisation of the model to the population, all patients were included and no specific exclusion criteria for stroke patients.

### Data from Riksstroke for external validation

We then used data from the Swedish stroke register (Riksstroke) for external validation of the developed models. Riksstroke is a national quality register, monitoring stroke care and outcome in adult patients. Details about the register and included variables have been published previously[11] and the forms can be found on Riksstroke's webpage (https://www.riksstroke.org/forms/).

In short, all hospitals admitting acute stroke (ischaemic, primary intracerebral haemorrhage, or unspecified type of stroke) patients in Sweden participate (currently 72 hospitals). Riksstroke is estimated to include more than 90% of all acute stroke patients treated in hospital. The register includes information from the entire chain of stroke care, including primary prevention, acute management, rehabilitation, secondary prevention, and family and community support. The dataset for external validation included patients who experienced stroke between 1 January 2015 and 31 December 2020.

### Availability of data and materials

The data that support the findings of this study are available from SSNAP (www.strokeaudit.org) and Riksstroke (www.riksstroke.org) but restrictions apply to the availability of these data, which were used under license for the current study, and so are not publicly available. Data are, however, available from the SSNAP and Riksstroke on reasonable request and with permission of HQIP and Riksstroke, respectively.

### Study outcome

For models developed with data from the SSNAP registry, the outcome was all-cause 30-day in-hospital mortality poststroke. All patients had in-hospital status due to the collection procedure in SSNAP. Out-hospital deaths were not available for analysis.

For the external validation with data from Riksstroke, date of death was retrieved for all patients by individual linkage with the national Cause of Death register, managed by the National Board of Health and Welfare, using personal identification numbers. Hence, all 30-day mortality included both in-hospital and out-of-hospital deaths.

The developed models with SSNAP data were externally validated with both Riksstroke in-hospital deaths

and all deaths as outcomes. Performances for in-hospital death were compared between SSNAP and Riksstroke. Comparisons between all deaths and in-hospital death in Riksstroke were conducted as a sensitivity analysis.

## Predictor variables

In total, 65 variables were available from SSNAP. Details on how each risk predictor was measured have been presented in a previous study for variables in SSNAP[7] and a previous study for variables in Riksstroke.[11] Subsequently, 13 variables were used to build and validate prediction models of 30-day mortality. The variable selection was done based on the clinical importance, literature review, and availability in both the SSNAP and Riksstroke registries. Different clinical covariates for prediction included age (band by 5), sex, inpatient at time of stroke, hour of admission, day of week of admission, comorbidities such as hypertension, atrial fibrillation (AF), diabetes, previous stroke or transient ischaemic attack (TIA) and anticoagulation prior to stroke if AF, prestroke modified Rankin Scale (mRS), loss of consciousness and type of stroke (ischaemic or primary intracerebral haemorrhage). The National Institutes of Health Stroke Scale (NIHSS) was not used as a variable/predictor due to the high missingness in Riksstroke (42.9%). Bray *et al.* 2014 showed that the level of consciousness component of the NIHSS provides a good approximation of the full NIHSS for mortality prediction.[6]

Two predictors were recorded differently in SSNAP and Riksstroke, that is, level of consciousness and pre-stroke mRS. In SSNAP, level of consciousness was assessed based on the first item of the NIHSS which has four levels, that is, 0=alert, 1=not alert but arousable by minor stimulation, 2=not alert and requires repeated stimulation to attend and 3=unconscious. In Riksstroke, level of consciousness was recorded in three categories where alert corresponds to level of consciousness 0 in SSNAP, not alert to level of consciousness 1–2 in SSNAP, and unconscious level of consciousness 3 in SSNAP. To obtain 4 categories in Riksstroke, a total NIHSS score≤13 vs >13 was used to split the mid category (not alert). The cut-off was chosen based on the NIHSS score distribution in Riksstroke and previous studies showing good discrimination properties for 30-day stroke mortality.[12] In Riksstroke prestroke mRS was estimated using a previously developed algorithm[13] which has four categories with the first category corresponding to 0–2 and the next three categories corresponding to 3, 4, 5 in the preStroke mRS in SSNAP. The first category was imputed to be 0 in Riksstroke for the external validation.

## Missing data

Missing data were handled using different methods according to assumptions of missing mechanism after consulting the SSNAP team and clinicians. Variables with more than 80% missingness were discarded. For categorical variables with missing by design/not applicable assumption, missing values were added as a new category.

Missing indicator was used for missing by design/not applicable continuous variables. After these, multiple imputation with chained equations (MICEs)[14] were used to impute variables with missing at random assumption for the development dataset. While the missing mechanism is impossible to prove, MICE is seen as a better choice according to several references when comparing it with different imputation methods.[15] All available variables except for the discarded ones were used in MICE. Five datasets were imputed using MICE and aggregated into one using the median for SSNAP with all the variables available in the dataset and Riksstroke dataset. Details for handling missing data in SSNAP were presented in a previous study.[7]

## Model building and evaluation

After variable selection and imputation of missing values, the variables were preprocessed. Details of preprocessing and coding of variables were presented in a previous study[7] and in the GitHub repository (https://github.com/WenjuanW/Risk_Prediction_of_Post-Stroke_30-day_Mortality_SSNAP_and_RiksStroke). Descriptive statistics were used to compare the characteristic of death/alive at 30-day across the entire datasets and between SSNAP and Riksstroke.

Models were developed/trained with data from SSNAP using 80% randomly selected patients from 2013 to 2018, validated on the remaining 20% data from 2013 to 2018, and temporally validated on data from 2019. Subsequently, models were externally validated using data from Riksstroke from between 1 January 2015 to 31 December 2020.

Due to the lack of final models from other ML studies and lack of certain variables for risk scores (PLAN score[8]: cancer, Iscore[9]: cancer, renal dialysis, stroke scale score, non-lacunar and abnormal glucose level), we could not externally validate existing ML models and risk scores. In previous studies comparing the performance of LR models with ML models, models developed using XGBoost generally performed the best.[7] Thus, new models were built with LR, LR with elastic net[16] with interaction terms, and XGBoost[17] using the mutual risk predictors that were available in both countries in order to externally validate the models. Best hyperparameters (a parameter that is predefined by the user to control the learning process) were selected on the development set with grid search or random search and cross-validation. Detailed hyperparameter tuning strategy was presented in a repository on Github that was built for this study (https://github.com/WenjuanW/Risk_Prediction_of_Post-Stroke_30-day_Mortality_SSNAP_and_RiksStroke).

Brier score[18] was used as an overall summative measure of predictive performance. Discrimination was measured by area under the receiver operating characteristic curve (AUC). Calibration was visually assessed with calibration plots[19] rather than Hosmer-Lemeshow (HL) test as HL test has a number of drawbacks, including limited power, poor interpretability and is not recommended.[19] Python

functions for calculating these measurements were presented in the Github repository.

Comparisons of AUCs and Brier scores within each set were conducted with Delong's test and paired t-test, respectively. To compare AUCs between the performances on SSNAP and Riksstroke, an independent group t-test was used for both AUCs and Brier scores with 500 bootstraps for each measurement.

Due to the low outcome rate, the dataset is imbalanced at predicting the outcomes. In this case, AUC cannot fully reflect the performances of the models on positive predictive rate (precision) and true positive rate (recall). Therefore, precision and recall were assessed via area under the precision-recall curves and the precision-recall curves.

Clinical utility was assessed with decision curve analysis[20] which graphically shows the net benefit obtained by applying the strategy of treating an individual if and only if predicted risk is larger than a threshold in function of the threshold probability. Threshold equals to 0 means treating all since all predicted risk will be larger than 0. Threshold equals to 1 means treating none since all predicted risk will be smaller than 1. All analyses were conducted using Python V.3.7.4.

As stroke type is a critical factor for outcomes of stroke patients, we subsequently performed a subgroup analysis to investigate the performance of the developed models for patients with different stroke types that is, infarction and primary intracerebral haemorrhage.

**Table 1** Patients characteristics in SSNAP and Riksstroke

| | SSNAP | | | Riksstroke (in-hospital mortality) | | |
|---|---|---|---|---|---|---|
| | Overall | Alive at 30 days | Dead at 30 days | Overall | Alive at 30 days | Dead at 30 days |
| No of cases (%) | 488 947 | 428 585 (87.6) | 60 362 (12.4) | 128 360 | 114 487 (89.2) | 13 873 (10.8) |
| Age 15–60 (%) | 76 941 (15.7) | 74 238 (17.3) | 2703 (4.5) | 15 586 (12.1) | 14 958 (13.1) | 628 (4.5) |
| Age 61–70 (%) | 84 484 (17.3) | 79 109 (18.5) | 5375 (8.9) | 23 186 (18.1) | 21 782 (19.0) | 1404 (10.1) |
| Age 71–80 (%) | 136 728 (28.0) | 122 393 (28.6) | 14 335 (23.7) | 41 396 (32.3) | 37 689 (32.9) | 3707 (26.7) |
| Age 81+ (%) | 190 794 (39.0) | 152 845 (35.7) | 37 949 (62.9) | 48 192 (37.5) | 40 058 (35.0) | 8134 (58.6) |
| Male (%) | 249 291 (51.0) | 223 849 (52.2) | 25 442 (42.1) | 68 985 (53.7) | 62 428 (54.5) | 6557 (47.3) |
| Hypertension (%) | 264 806 (54.2) | 232 505 (54.2) | 32 301 (53.5) | 80 824 (63.0) | 71 481 (62.4) | 9343 (67.4) |
| AF (%) | 96 354 (19.7) | 76 417 (17.8) | 19 937 (33.0) | 37 090 (28.9) | 31 057 (27.1) | 6033 (43.5) |
| Diabetes (%) | 102 324 (20.9) | 90 104 (21.0) | 12 220 (20.2) | 28 722 (22.4) | 25 589 (22.4) | 3133 (22.6) |
| Previous stroke or TIA (%) | 129 462 (26.5) | 112 358 (26.2) | 17 104 (28.3) | 35 145 (27.4) | 30 757 (26.9) | 4388 (31.6) |
| NIHSS at arrival (median (IQR)) | 4.00 (2.0, 10.0) | 4.00 (2.0, 8.0) | 17.00 (7.0, 23.0) | 3.00 (1.0, 8.0) | 3.00 (1.0, 7.0) | 16.00 (9.0, 23.0) |
| Level of consciousness (%)* | | | | | | |
| 0 (alert) | 391 065 (80.0) | 379 085 (88.5) | 11 980 (19.8) | 106 474 (83.0) | 101 706 (88.8) | 4768 (34.4) |
| 1 (not alert) | 41 916 (8.6) | 32 715 (7.6) | 9201 (15.2) | 14 067 (11.0) | 9459 (8.3) | 4608 (33.2) |
| 2 (not alert) | 22 635 (4.6) | 11 790 (2.8) | 10 845 (18.0) | | | |
| 3 (unconscious) | 33 331 (6.8) | 4995 (1.2) | 28 336 (46.9) | 5930 (4.6) | 1759 (1.5) | 4171 (30.1) |
| Pre-stroke mRS (%) | | | | | | |
| 0 | 266 409 (54.5) | 245 979 (57.4) | 20 430 (33.8) | 87 593 (68.2) | 81 871 (71.5) | 5722 (41.3) |
| 1 | 75 661 (15.5) | 66 940 (15.6) | 8721 (14.4) | | | |
| 2 | 50 795 (10.4) | 42 592 (9.9) | 8203 (13.6) | | | |
| 3 | 56 994 (11.7) | 44 601 (10.4) | 12 393 (20.5) | 18 942 (14.8) | 16 563 (14.5) | 2379 (17.2) |
| 4 | 30 276 (6.2) | 22 339 (5.2) | 7937 (13.1) | 11 763 (9.2) | 9483 (8.3) | 2280 (16.4) |
| 5 | 8812 (1.8) | 6134 (1.4) | 2678 (4.4) | 3623 (2.8) | 2720 (2.4) | 903 (6.5) |
| Haemorrhage (%) | 55 758 (11.5) | 39 472 (9.3) | 16 286 (27.2) | 16 757 (13.1) | 12 074 (10.6) | 4683 (33.8) |

*Level of consciousness is measured differently between SSNAP. In SSNAP, level of consciousness has four levels (0=alert, 1=not alert but arousable by minor stimulation, 2=not alert and requires repeated stimulation to attend and 3=unconscious). In Riksstroke, level of consciousness has three levels (0=alert, 1, 2=not alert, 3=unconscious).

AF, atrial fibrillation; mRS, modified Rankin Scale; NIHSS, National Institutes of Health Stroke Scale; SSNAP, Sentinel Stroke National Audit Programme; TIA, transient ischaemic attack.

## Sensitivity analysis

External validation on all 30-day mortality (i.e., including deaths outside of hospital) in Riksstroke was conducted as a sensitivity analysis to check the impact of using only in-hospital mortality for developing the models. Another sensitivity analysis was done using mRS=1 as the level representing 0–2 in previous stroke mRS variable in Riksstroke.

## Patient and public involvement

Neither the patients nor the public were involved in this study.

## RESULTS
## Participants

The characteristics of 488 947 patients from SSNAP and 128 360 patients from Riksstroke were reported in table 1. In total, 60 362 (12.4%) patients from SSNAP and 13 873 (10.8%) from Riksstroke died within 30 days in hospital and (13.1% for all mortality within 30 days). In both SSNAP and Riksstroke, patients who died within 30 days were older, and were more likely to have cardiovascular risk factors, impaired functional status prior to stroke, haemorrhagic stroke or being unconscious on arrival to hospital (table 1).

The patient casemix in SSNAP and Riksstroke was similar with respect to age distribution. Riksstroke had a high percentage of males for patients who died at 30 days. A higher frequency in Riksstroke compared with SSNAP was observed in hypertension (63.0% vs 54.2%), AF (28.9% vs 19.7%), diabetes (22.4% bs 20.9%), previous stroke or TIA (27.4% vs 26.5%) and haemorrhage (13.1% vs 11.5%). The difference in these risk factors is even larger for the patients who died with 30 days after stroke with hypertension (67.4% vs 53.5%), AF (43.5% vs 33.0%), diabetes (22.6% bs 20.2%), previous stroke or TIA (31.6% vs 28.3%) and haemorrhagic stroke (33.8% vs 27.2%). A lower percentage of high level of consciousness (level 3) was observed in Riksstroke compared with SSNAP (4.6% vs 6.8%). On the contrary, a higher percentage of high levels of previous stroke mRS was found in Riksstroke compared with SSNAP (2.82% vs 1.8% for level 5, 9.2% vs 6.2% for level 4, and 14.8% vs 11.7% for level 3).

Summary of missing percentages and imputation methods for all predictors were presented in online supplemental table A. In SSNAP, only the type of stroke predictor had missing values of 0.7% and the missing values were taken as a new category. In Riksstroke, prestroke mRS had 5.0% missing values and level of consciousness had 1.5% missing values. Missing percentages for comorbidity variables were small (less than 0.5%).

## Model specification and performance

General characteristics for the development, validation and temporal validation set on all candidate variables are presented in previous study.[7] Specifications of the trained models and explanations on how to use them can be found in the repository on Github (https://github.com/WenjuanW/Risk_Prediction_of_Post-Stroke_30-day_Mortality_SSNAP_and_RiksStroke). All models performed similarly on training set and validation set indicating the models were not overfitting (online supplemental table B) and calibrated well (online supplemental figure A).

For the performances on 2019 temporal validation with SSNAP data (table 2), XGBoost obtained the lowest Brier score of 0.077 (95% CI 0.075 to 0.079) and the highest AUC of 0.852 (95% CI 0.846 to 0.858) compared with LR with elastic net and interaction terms (Brier score: 0.078 (95% CI 0.076 to 0.080) and AUC 0.850 (95% CI 0.844 to 0.856)) and LR (Brier score: 0.078 (95% CI 0.076 to 0.080) and AUC 0.847 (95% CI 0.841 to 0.853)). The differences between the models were all significant (p<0.001) even though small.

For the performances on external validation with Riksstroke data (table 2) with in-hospital mortality, XGBoost achieved the lowest Brier score (0.066 (95% CI 0.065 to 0.067) vs 0.067 (95% CI 0.066 to 0.068) from LR with elastic net and interaction terms vs 0.066 (95% CI 0.065 to 0.067) from LR, p<0.001). In terms of AUC, XGBoost achieved the slightly better performance compared with LR with elastic net and interaction terms (0.861 (95% CI 0.858 to 0.865) vs 0.862 (95% CI 0.859 to 0.866), p=0.008) while better than LR (0.861 (95% CI 0.857 to 0.864), p<0.001). The performances of the models on Riksstroke data were slightly better than on SSNAP temporal validation data. The AUC between in-hospital and all-mortality in RiksStroke was similar (online supplemental table C)

**Table 2** Brier score, AUC with 95% CI for 2019 temporal validation in SSNAP and Riksstroke external validation with in-hospital mortality

| | Brier score (95% CI) | | AUC (95% CI) | |
|---|---|---|---|---|
| | SSNAP | Riksstroke | SSNAP | Riksstroke |
| LR | 0.078 (0.076 to 0.080) | 0.066 (0.065 to 0.067) | 0.847 (0.841 to 0.853) | 0.861 (0.857 to 0.864) |
| LR with elastic net and interaction terms | 0.078 (0.076 to 0.080) | 0.067 (0.066 to 0.068) | 0.850 (0.844 to 0.856) | 0.862 (0.859 to 0.866) |
| XGBoost | 0.077 (0.075 to 0.079) | 0.066 (0.065 to 0.067) | 0.852 (0.846 to 0.858) | 0.861 (0.858 to 0.865) |

AUC, area under the curve; LR, logistic regression; SSNAP, Sentinel Stroke National Audit Programme; XGBoost, eXtreme Gradient Boosting.

From the precision-recall curves (figure 1), the models were better at predicting precision on the Riksstroke data compared with on SSNAP data. Calibration plots in figure 1 shows good calibration for the 2019 temporal validation data in SSNAP with slightly underestimation of the risks for patients with medium risks (0.2–0.6). The models overestimated the risks for in-hospital mortality in Riksstroke for about 0.1 for medium risks patients while the overestimation was smaller for the all mortality predictions in Riksstroke. Decision curves on SSNAP 2019 temporal validation set and Riksstroke external validation set showed all models gained similar net benefits[20] (online supplemental figure B).

For different stroke types, all models performed better on haemorrhagic patients than infarction patients on both SSNAP data (figure 2) and Riksstroke data (figure 3 for in-hospital mortality and (online supplemental figure C) for all-mortality) in terms of AUC and area under precision-recall curves. Calibrations were similar between different stroke types (figures 2 and 3 and online supplemental figure C).

According to the feature importance calculated from the XGBoost model (online supplemental figure D), level of consciousness, type of stroke, age, prestroke mRS and AF was the most important features in making the predictions which were in line with the literature. The coefficients from LR model (online supplemental table D) also confirmed the importance of these variables.

Through the sensitivity analysis of using prestroke mRS=1 as the level representing 0–2 in previous stroke mRS variable in Riksstroke, we found no impact on the performances.

## DISCUSSION

This is the first cross-national study to develop and externally validate an ML-based risk prediction model for 30-day mortality after stroke that is suitable for use in care quality registers. Mortality is an important outcome after stroke and widely reported by stroke quality initiatives as a measure of quality and safety of acute stroke care, including in the Riksstroke and SSNAP national quality registers used in this study. The results of the internal validation in the UK and the external validation in Sweden were similar and together provide evidence that the model was accurate and generalisable across countries and data sources. This is promising for developing a single model that can be used to benchmark different hospitals/organisations in different countries and make fairer comparisons of stroke outcomes across countries. But note that the UK and Sweden both have similar European patient populations, and there are many similarities between health systems. It is not as certain whether this model would be generalisable to a nation with a much more different patient population or care delivery system.

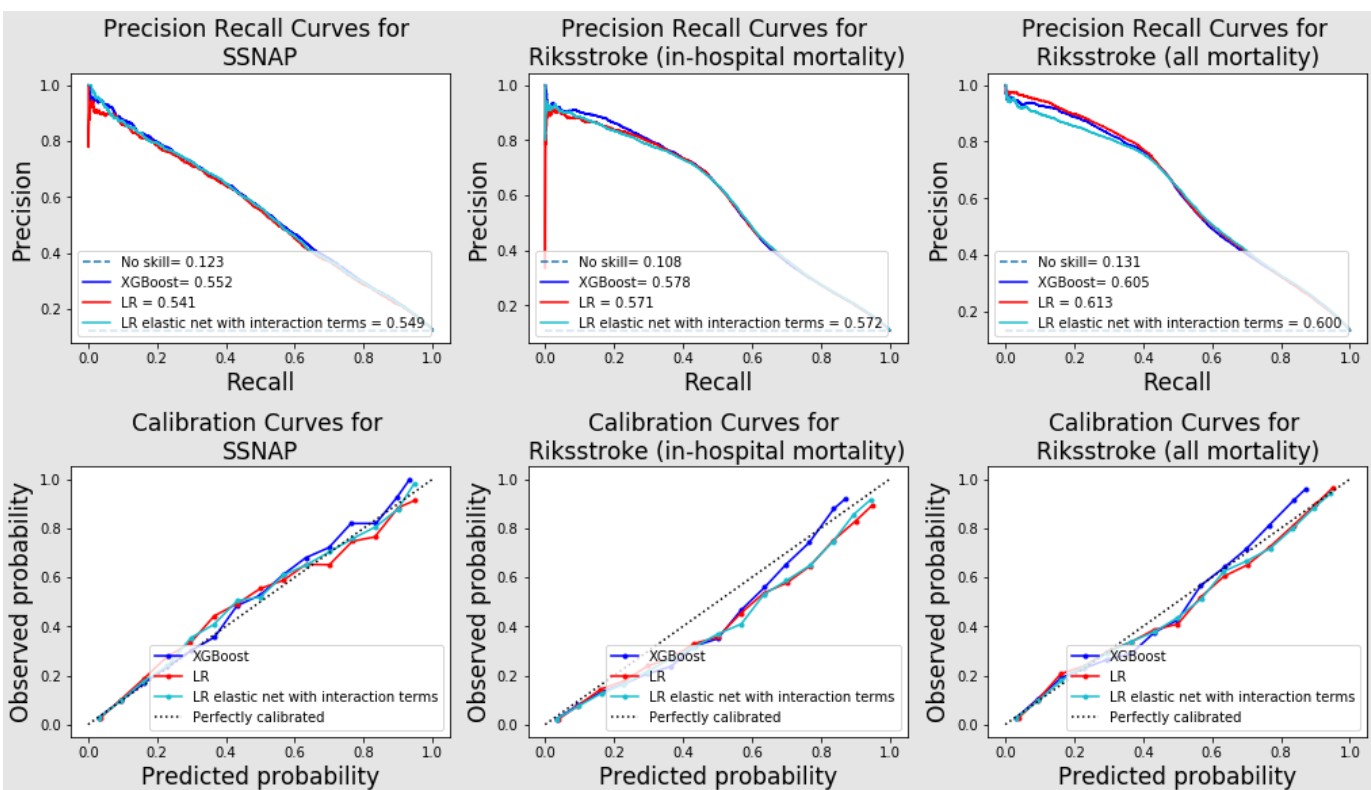

**Figure 1** Precision-recall curves (above) and calibration curves (below) for all models on temporal validation of SSNAP cohort (left) and external validation using Riksstroke with in-hospital mortality (middle) and Riksstroke with all mortality (right). SSNAP, Sentinel Stroke National Audit Programme.

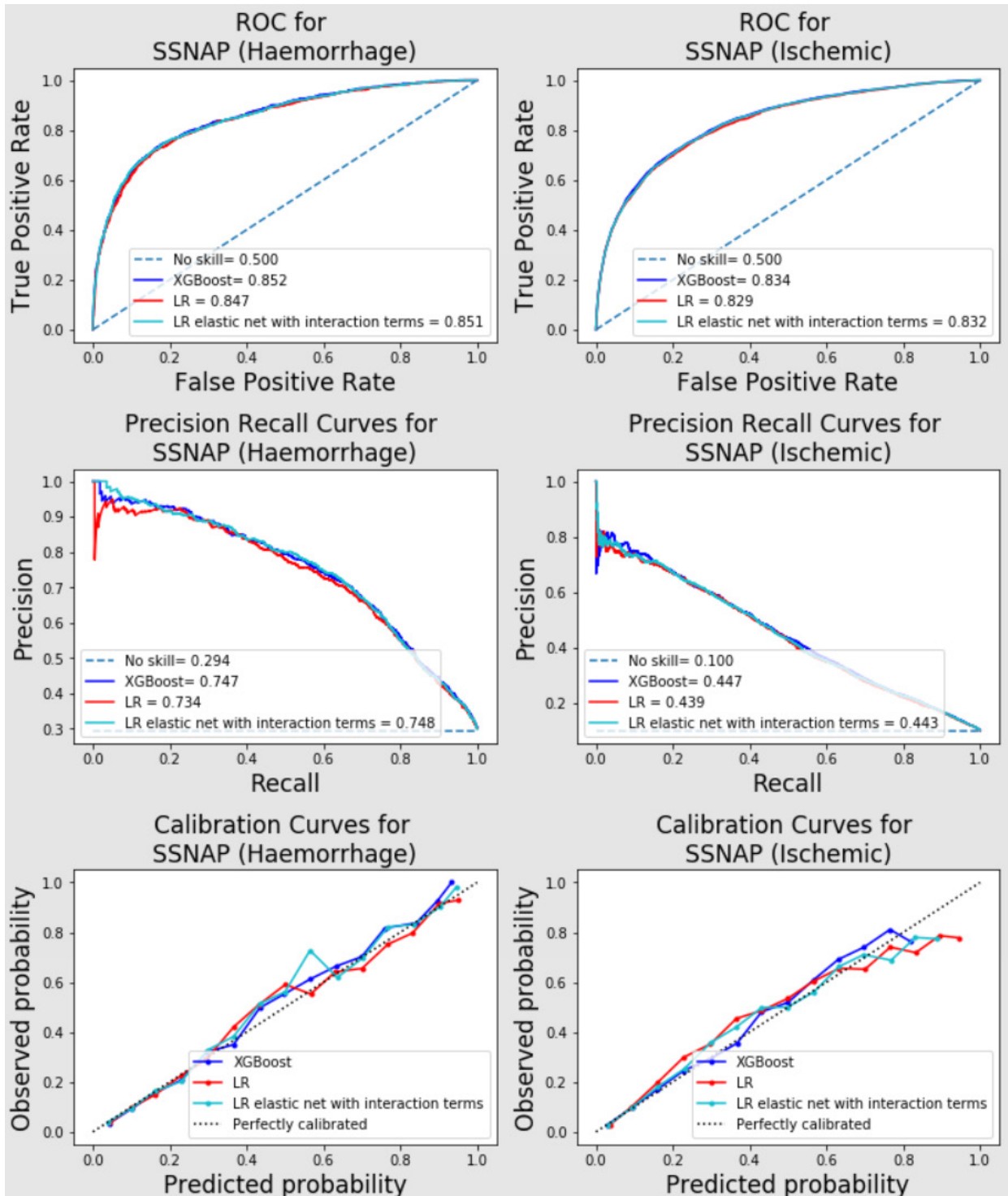

**Figure 2** AUC-ROC, PR curve and calibration curve for in-hospital mortalities (haemorrhage (left) vs Ischaemic (right) patients) in SSNAP temporal validation set. AUC, area under the curve; ROC, receiver operating characteristic curve; LR, logistic regression; PR, precision recall; SNNAP, Sentinel Stroke National Audit Programme; XGBoost, eXtreme Gradient Boosting.

Externally validated prediction models can be valuable tools in benchmarking healthcare outcomes across different hospitals[1] or countries and thereby contributing to improved clinical practice. This can be achieved by comparing the standardised mortality ratio (SMR)[21] which is calculated using the observed mortality rate and the predicted risk of mortality. A higher than expected SMR may be an indicator of poorer quality care, and there is strong evidence that higher quality acute stroke care (such as stroke unit based care[22]) reduces the risk of dying after stroke. The validity of this approach is, however, dependent on being able to make an accurate prediction of mortality, and hence it is important that the models

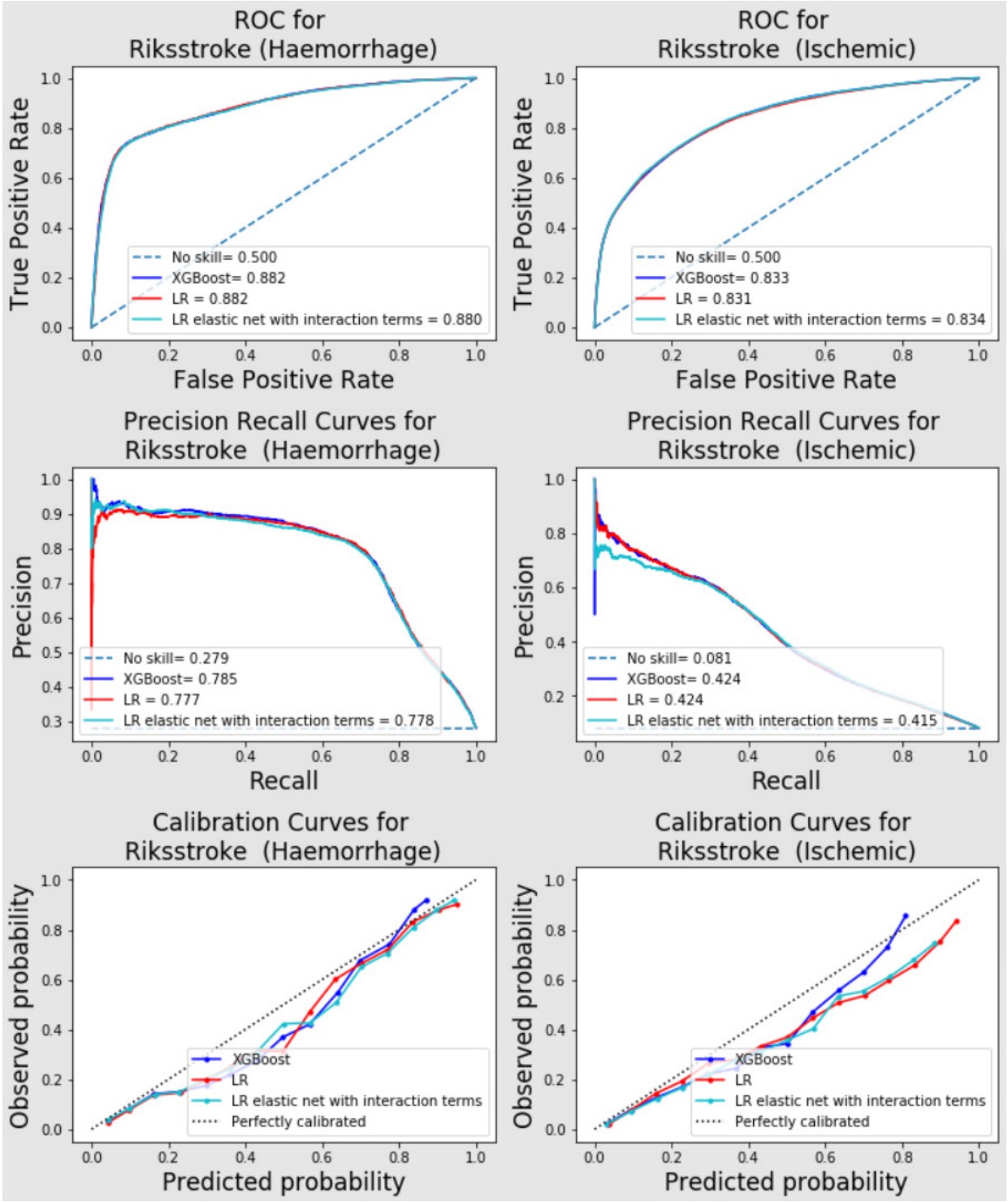

**Figure 3** AUC-ROC, PR curve and calibration curve for in-hospital mortalities (haemorrhage (left) vs Ischaemic (right) patients) in Riksstroke. AUC, area under the curve; ROC, receiver operating characteristic curve; LR, logistic regression; PR, precision recall; XGBoost, eXtreme Gradient Boosting.

used to generate these predictions are as accurate as possible and have been thoroughly validated. Another use case for validated models could be identifying patients with high risk for intervention. Patients at high risk for 30-day mortality should be considered for intensive outpatient interventions, and further studied to see if this reduces their risk.

We found that the performance of the model in external validation with Riksstroke data was in fact slightly better than in the temporal validation using data from SSNAP. This may be because the models performed more accurately in patients with haemorrhagic stroke and Riksstroke patients had a higher prevalence of haemorrhage patients. Also, the model was calibrated better for

all mortality than in-hospital mortality in Riksstroke. The reason for this is unclear, but one possible explanation could be that the in-hospital mortality rate in Riksstroke (10.8%) was lower than in SSNAP (12.4%) so the prediction model built from SSNAP overestimated the mortality rate in Riksstroke.

Although we could not externally validate the models with the existing stroke prognostic scores, our models showed performance accuracy similar to that of these scores in predicting in-hospital mortality. One study[23] has assessed the performance of 6 stroke prognostic scores in 4237 acute ischaemic stroke patients hospitalised at a Japanese stroke centre between 2012 and 2017 including existing risk score PLAN[8] with AUC (0.87 (95% CI 0.85 to 0.90))[23] and IScore with AUC (0.88 (95% CI 0.86 to 0.91)).[23] The performances in the Japanese cohort were higher compared with the original studies (PLAN: AUC 0.79, IScore: AUC 0.79–0.85) but the findings may not generalisable to other stroke populations. There have been many ML models built for predicting stroke mortality.[3 24] Unfortunately, we are unable to externally validate these as the final models were not published. The objectives for these studies were to obtain accurate prognostication of stroke for helping with therapy and rehabilitation planning. In this study, we aimed to have models that were developed in one nation and externally validated in another so the models could be used as benchmarks for evaluating quality of stroke care.

Regarding the comparisons between ML and regression based models, our study also confirmed that there is only a small performance benefit of ML over carefully developed LR models for clinical prediction models, which was consistent with prior literature.[5 7] While the AUC between models may have been statistically significant, the differences were small and unlikely clinically significant. In this case, where a limited number of predicting features were included, LR performed similarly to ML and ML might not add much discriminatory value compared with LR. However, ML has outperformed LR in some other settings.[3] The role of ML in quality improvement has not been extensively explored. This study suggests that ML could have a role as part of the analysis and feedback of data about healthcare quality in specific situations without pre-specified model assumptions and restrictions in the number of included variables.

### Strengths and limitations

The models were built and externally validated with very large and comprehensive stroke datasets (including data on 488 497 patients for development and validation and 128 360 patients for the external validation). Both data sources include a nearly complete population of hospitalised stroke in England and Wales (SSNAP) and Sweden (Riksstroke) reducing the risk of bias.

The models were built with robust approaches and reported according to the TRIPOD reporting guidelines with several terms adjusted for ML studies such as hyperparameter turning and final model presentation. The

models were published and have been made freely available for download for future external validation and use (GitHub repository: https://github.com/WenjuanW/Risk_Prediction_of_Post-Stroke_30-day_Mortality_SSNAP_and_RiksStroke).

The main limitations are: (1) Death outcomes were limited to inpatient mortality, and it was not possible to ascertain deaths occurring outside hospital within 30 days for SSNAP data. Hence, the rates of 30-day mortality are likely underestimated and imprecise. But from the sensitivity analysis using all mortality from Riksstroke data, the rate of out-hospital mortality is small (2.3%) and the performance of external validation was good for both in-hospital mortality and all mortality. (2) NIHSS as a main risk predictor for mortality was not available in Riksstroke. Level of consciousness was used instead and according to previous studies,[6 6] the level of consciousness component of the NIHSS provides a good approximation of the full NIHSS for mortality prediction in settings where the ability to record the full NIHSS on admission is limited. (3) Some other risk factors are not available such as imaging, congestive heart failure etc. (4) The variables were limited to the ones that are available in both SSNAP and Riksstroke. However, all variables are generally available or can be derived from available variables in stroke registries which can benefit from the models developed in this study. (5) As can be expected, there were some discrepancies in how the included variables were recorded in the two registers. Well-constructed methods were applied to handle the discrepancies and further sensitivity analysis was conducted. (6) Only XGBoost was used for developing the model from ML-based methods but as we know that generally ML methods are not outperforming statistical methods with structured data, more ML methods would not be likely to add more information due to the limited number of variables.

### CONCLUSIONS

We have developed and externally validated a ML based model to predict 30-day mortality after stroke, a widely collected and reported measure of stroke quality. The development of these types of generalisable models are important in allowing for unbiased comparisons of stroke outcomes across countries, and would be well suited to support efforts to measure and compare stroke outcomes as part of quality improvement initiatives.

**Contributors** Study design: WW, ME and BDB. Literature review: WW, JAO, ME and BDB. Figures: WW and JAO. Data cleaning and preprocessing: WW and JAO. Model developing and internal validation (UK): WW and BDB. External validation (Sweden): JAO and ME. Data interpretation: WW, JAO, ME, BDB, CDW and VC. Writing—original draft: WW and JAO. Writing—review and editing: WW, BDB, ME, JAO, CW and VC. Supervision: BDB (UK), ME (Sweden) Funding acquisition (UK): BDB, CDW and VC. Project administration (UK): WW, BDB and VC. Guarantor: WW, BDB, JAO, ME.

**Funding** CDW, BDB, VC hold the award from the Health Foundation. Award Number is 553013. CDW and VC acknowledge support from the National Institute for Health Research (NIHR) Biomedical Research Centre (BRC) based at Guy's and St Thomas' National Health Service (NHS) Foundation Trust and King's College London, and the NIHR Collaboration for Leadership in Applied Health Research and Care (ARC) South

London at King's College Hospital NHS Foundation Trust. VC was also supported by EPSRC grant EP/P010105/2, CONSULT: Collaborative Mobile Decision Support for Managing Multiple Morbidities and EP/X030628/1, King's Health Partners Digital Health Hub.

**Disclaimer** The views expressed are those of the authors and not necessarily those of the NHS, the BRC or ARC.

**Competing interests** None declared.

**Patient and public involvement** Patients and/or the public were not involved in the design, or conduct, or reporting, or dissemination plans of this research.

**Patient consent for publication** Not applicable.

**Ethics approval** SSNAP has approval from the Clinical Advisory Group of the NHS Health Research Authority to collect patient-level data under section 251 of the NHS Act 2006. No additional ethical review was sought. Informed consent was not sought for the present study because data analyses were carried out using fully anonymised datasets from SSNAP. The external validation using Riksstroke was approved by the Swedish Ethical Review Authority (reference no. 2021-06152-01). In accordance with the Personal Data Act (Swedish law No. SFS 1998:204), no informed consent is needed to collect data from medical charts and inpatient records for quality registers.

**Provenance and peer review** Not commissioned; externally peer reviewed.

**Data availability statement** Data may be obtained from a third party and are not publicly available. The data that support the findings of this study are available from SSNAP (www.strokeaudit.org) and Riksstroke (www.riksstroke.org) but restrictions apply to the availability of these data, which were used under licence for the current study, and so are not publicly available. Data are, however, available from the SSNAP and Riksstroke on reasonable request and with permission of HQIP and Riksstroke, respectively.

**ORCID iDs**
Wenjuan Wang http://orcid.org/0000-0002-1879-7332
Josline A Otieno http://orcid.org/0009-0006-1030-4237
Marie Eriksson http://orcid.org/0000-0003-3298-1555
Charles D Wolfe http://orcid.org/0000-0001-8264-0981

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
