## [Reviewer comments · BMJ Open]

ARTICLE DETAILS

TITLE (PROVISIONAL)	Developing and externally validating a machine learning risk prediction model for 30-day mortality after stroke using national stroke registers in the UK and Sweden
AUTHORS	Wang , Wenjuan; Otieno, Josline; Eriksson, Marie; Wolfe, Charles; Curcin, Vasa; Bray, Benjamin

VERSION 1 – REVIEW

REVIEWER	Huo, Zepeng Stanford University, Biomedical Informatics
REVIEW RETURNED	14-Feb-2023

GENERAL COMMENTS	This paper presents a cross-country (UK, Sweden) study for machine learning methods on risk prediction, specifically mortality of stroke patients in hospital setting. Overall the paper presents a clear idea and the experiments that support it are relatively clear. However there are some minor issues on the paper: - They stated that PLAN and IScore were not compared due to lack of variables. But can they conduct the experiments on the available variables to see how much performance difference is there. Perhaps this can give insight on the importance on those variables.- They used Multiple Imputation with Chained Equations (MICE) to deal with the missing at random assumption. But isn't the technique assuming some correlations across the channels of the missing and non-missing data to impute? Can they elaborate the intuition for using this technique?- Brier score is both for calibration and ranking purposes. They mainly used calibration plot to show the calibration performance but what's the analysis on brier score? In addition, did they consider brier skill score? Since brier score sometimes gives over-optimistic results- For figure1 (which is missing the title and caption, so I assume it's the first figure), why does in-hospital mortality have the worst performance on calibration plot across the methods? What's the likely cause?- Overall, the figures are in low resolution so some numbers are hard to read
--

REVIEWER	Reed, Grant Cleveland Clinic
REVIEW RETURNED	20-Mar-2023

GENERAL COMMENTS

It is my pleasure to review this article by Wang, et al. In this study, the authors use data from the UK SSNAP registry to create a stroke mortality risk score, and the Swedish Riks-Stroke database as a validation cohort, using logistic regression (LR) and machine learning (ML) techniques. The authors are able to achieve strong predictive ability for the prediction of 30-day mortality using both LR and the ML XGBoost model.

There are several strengths of the study. The authors evoke a very large sample size of 488,497 patients in the UK cohort and 128,360 patients in the Swedish validation group. The event rate is frequent enough at 10-13% in both studies, so the study is well powered to assess the outcome of interest.

Specific suggestions the authors may wish to consider to improve the manuscript:

- 1- In general, the manuscript is well written. However, there are several instances of run-on sentences, which could be edited for clarity. For instance, in the abstract, the Conclusion is a single, long sentence which is confusing as written.
- 2- The authors acknowledge the major limitations, including the inability to ascertain out of hospital deaths within 30-days in the SSNAP data. It is important to emphasize that the rates of 30-day mortality are likely underestimated, and imprecise.
- 3- While it is true NIHSS is a strong predictor of stroke mortality, and the authors are unable to include this as it is not available in the Riksstroke data, the authors would be well served by commenting on whether including NIHSS improved the AUC of the model, even if it could not be validated. If it did not, the authors could reasonably conclude that although NIHSS was not available in the validation cohort and thus the final model did not include it, given the robustness of the model they created without NIHSS as a variable, NIHSS would not have added further discriminatory ability and thus was not necessary anyway.
- 4- My interpretation of the data as that although the ML model predicted mortality best, the LR model did nearly as well. While the AUC between models may have been statistically significant, the differences were small and unlikely clinically significant. LR models tend to lend themselves toward forming risk scores better than ML models do. Thus, it is important to state that although ML did have good predictive value, LR actually performed similarly, and it is not likely to have added much discriminatory value over LR alone.
- 5- It would be helpful to provide perspective in the Discussion on how the models created perform in comparison to existing stroke mortality risk scores.
- 6- The authors make comments about how creation of this model may improve patient quality, though should comment more about how. One potential application of this would be to create a clinical risk score based on these data, which could be made available online. Patients at high risk for 30-day mortality should be considered for intensive outpatient interventions, and further studied to see if this reduces their risk.
- 7- The authors also comment that models developed in one nation may not perform well in another nation, and use this study as an opportunity to state that this is not the case. It is important to note that UK and Sweden both have similar European patient populations, and there are many similarities between health systems. It is not as certain whether this model would be generalizable to a nation with a much more different patient population or care delivery system (i.e. China).

	8- While I do follow the methodology and agree it seems reasonable and well executed, I recommend a statistical reviewer analyze the methodology closely given the complexity of the techniques used.
--	---

REVIEWER	Fontoura Solla, Davi University of São Paulo Department of Neurosciences and Behaviour Sciences
REVIEW RETURNED	05-May-2023

GENERAL COMMENTS	The authors are to be commended for the study. Although their previous paper (https://pubmed.ncbi.nlm.nih.gov/36008776/) didn't show a "clinically meaningful" superiority of Machine Learning over Logistic Regression based statistics for stroke risk prediction, ML is indeed becoming a more available and useful tool, so the matter is valuable. The statistical analyses of the study were well performed. The major limitations are data availability and data ascertainment, as well as in any other study with this kind of dataset. My biggest concern is that I didn't perceive any clear differential of this study over other stroke ML prediction risk initiatives (https://pubmed.ncbi.nlm.nih.gov/36580703/), except for being cross-national (but still within the same continent - and similar ethnicity). The comparison with other stroke ML prediction risk initiatives could be better explored in the discussion section.
--

VERSION 1 – AUTHOR RESPONSE

Reviewer: 1

Dr. Zepeng Huo, Stanford University

Comments to the Author:

This paper presents a cross-country (UK, Sweden) study for machine learning methods on risk prediction, specifically mortality of stroke patients in hospital setting.

Overall the paper presents a clear idea and the experiments that support it are relatively clear.

However there are some minor issues on the paper:

- They stated that PLAN and IScore were not compared due to lack of variables. But can they conduct the experiments on the available variables to see how much performance difference is there. Perhaps this can give insight on the importance on those variables.

Response: Thanks for the suggestion. To gain some insight, we have added the performance of PLAN and IScore in Discussion for comparison. We agree that a performance can still be obtained by setting the missing variable with a chosen value (otherwise the prediction cannot be obtained without the missing variable). But the performance of the score will be different using different values. We might not be certain of the importance of these variables due to lack of ground truth (the missing variables).

- They used Multiple Imputation with Chained Equations (MICE) to deal with the missing at random

assumption. But isn't the technique assuming some correlations across the channels of the missing and non-missing data to impute? Can they elaborate the intuition for using this technique?

Response: Thanks for the question. MICE is a widely used method and generally considered fairly robust to different patterns of data where missing at random (MAR) is assumed. We have information on a wide set of covariates and outcome, including e.g. level of consciousness and mortality so the deviation from the MAR assumption might be minor. While the missing mechanism is impossible to prove, MICE is seen as a better choice according to several references when comparing it with different imputation methods [Baneshi et al. 2012]. We have added more details on MICE in the Methods section.

[Baneshi et al. 2012]: Baneshi MR, Talei AR. Does the missing data imputation method affect the composition and performance of prognostic models? Iran Red Crescent Med J. 2012 Jan;14(1):31-6. Epub 2012 Jan 1. PMID: 22737551; PMCID: PMC3372019.

- Brier score is both for calibration and ranking purposes. They mainly used calibration plot to show the calibration performance but what's the analysis on brier score? In addition, did they consider brier skill score? Since brier score sometimes gives over-optimistic results

Response: Thanks for the suggestion. We think the same and presented Brier score (Table 2) as an overall score for the performance of discrimination and calibration. Then we used AUC as a score for discrimination and calibration plot for the performance of calibration. According to [some references], these are the recommended scores when comparing performances of predictive models. Brier skill score is the brier score of reference or baseline predictions which one seeks to improve on. Thus a brier skill score might not be the best for our purpose which is to compare several methods all together. We have conducted statistical tests to compare the significance of the differences between the scores from different methods.

- For figure1 (which is missing the title and caption, so I assume it's the first figure), why does in-hospital mortality have the worst performance on calibration plot across the methods? What's the likely cause?

Response: Thanks for the insightful question. The reason for this is unclear, but one possible explanation could be that the in-hospital mortality rate in Riksstroke (10.8%) was lower than in SSNAP (12.4%) so the prediction model built from SSNAP overestimated the mortality rate in Riksstroke. We have added this in discussion.

- Overall, the figures are in low resolution so some numbers are hard to read

Response: Thanks for the suggestion. We have updated the figures to have a better resolution.

Reviewer: 2

Dr. Grant Reed, Cleveland Clinic

Comments to the Author:

It is my pleasure to review this article by Wang, et al. In this study, the authors use data from the UK SSNAP registry to create a stroke mortality risk score, and the Swedish Riks-Stroke database as a validation cohort, using logistic regression (LR) and machine learning (ML) techniques. The authors are able to achieve strong predictive ability for the prediction of 30-day mortality using both LR and the ML XGBoost model.

There are several strengths of the study. The authors evoke a very large sample size of 488,497 patients in the UK cohort and 128,360 patients in the Swedish validation group. The event rate is frequent enough at 10-13% in both studies, so the study is well powered to assess the outcome of interest.

Response: Thanks to Dr. Grant Reed for the nice summary.

Specific suggestions the authors may wish to consider to improve the manuscript:

1- In general, the manuscript is well written. However, there are several instances of run-on sentences, which could be edited for clarity. For instance, in the abstract, the Conclusion is a single, long sentence which is confusing as written.

Response: Thanks for the suggestion. We have refined the manuscript and tried to be clearer.

2- The authors acknowledge the major limitations, including the inability to ascertain out of hospital deaths within 30-days in the SSNAP data. It is important to emphasize that the rates of 30-day mortality are likely underestimated, and imprecise.

Response: Thanks for the advice. We agree and have added this in Discussion.

3- While it is true NIHSS is a strong predictor of stroke mortality, and the authors are unable to include this as it is not available in the Riksstroke data, the authors would be well served by commenting on whether including NIHSS improved the AUC of the model, even if it could not be validated. If it did not, the authors could reasonably conclude that although NIHSS was not available in the validation cohort and thus the final model did not include it, given the robustness of the model they created without NIHSS as a variable, NIHSS would not have added further discriminatory ability and thus was not necessary anyway.

Response: Thanks for the suggestion. We agree and we have mentioned in methods that Bray et al. 2014 compared the predictive performance using NIHSS score and level of consciousness. The performances showed comparable predictive ability. Moreover, level of consciousness was used as a variable in the existing risk score PLAN.

[Bray et al. 2014]: Bray BD, Campbell J, Cloud GC, et al. Derivation and external validation of a case mix model for the standardized reporting of 30-day stroke mortality rates. Stroke. 2014;45(11):3374-3380. doi:10.1161/STROKEAHA.114.006451

4- My interpretation of the data as that although the ML model predicted mortality best, the LR model did nearly as well. While the AUC between models may have been statistically significant, the differences were small and unlikely clinically significant. LR models tend to lend themselves toward forming risk scores better than ML models do. Thus, it is important to state that although ML did have good predictive value, LR actually performed similarly, and it is not likely to have added much discriminatory value over LR alone.

Response: Thanks for the comments. We absolutely agree and have added this in discussion. In a previous study for comparing ML and LR in the SSNAP cohort, we had a similar conclusion.

5- It would be helpful to provide perspective in the Discussion on how the models created perform in comparison to existing stroke mortality risk scores.

Response: Thanks for the suggestion. We have added some comparisons to the existing stroke mortality risk scores in literature. We could not externally validate these risk scores due to the lack of final models from other ML studies and lack of certain variables for risk scores as we now mentioned in Methods.

6- The authors make comments about how creation of this model may improve patient quality, though should comment more about how. One potential application of this would be to create a clinical risk score based on these data, which could be made available online. Patients at high risk for 30-day mortality should be considered for intensive outpatient interventions, and further studied to see if this reduces their risk.

Response: Thanks for the advice. We agree and have added the suggested use case in Discussion. The final model has been published online and available to everyone to use which functions the same as a clinical score. The aim of the paper was to build a model for benchmarking the quality of stroke care in different countries. Using the expected mortality rate and the observed mortality rate to decide if the quality of care in one country is underperforming so actions could be taken to improve the quality of stroke care. We have clarified this in the Discussion section. The prediction model was not intended to be used as a tool for identifying high risk patients.

7- The authors also comment that models developed in one nation may not perform well in another nation, and use this study as an opportunity to state that this is not the case. It is important to note that UK and Sweden both have similar European patient populations, and there are many similarities between health systems. It is not as certain whether this model would be generalizable to a nation with a much more different patient population or care delivery system (i.e. China).

Response: Thanks for the comments. We agree and have added this to Discussion. Before use in a population other than training the models, additional validation is needed especially in health systems

with different populations or organisation of care systems. For this purpose, we have made our models publicly available.

8- While I do follow the methodology and agree it seems reasonable and well executed, I recommend a statistical reviewer analyze the methodology closely given the complexity of the techniques used.

Response: Thanks for the suggestion. Reviewer 3 mentioned that the statistical analyses of the study were well performed.

Reviewer: 3

Dr. Davi Fontoura Solla, University of São Paulo Department of Neurosciences and Behaviour Sciences

Comments to the Author:

The authors are to be commended for the study.

Although their previous paper

(<https://eur03.safelinks.protection.outlook.com/?url=https%3A%2F%2Fpubmed.ncbi.nlm.nih.gov%2F36008776%2F&data=05%7C01%7Cvasa.curcin%40kcl.ac.uk%7Cec49d280ec734d3e1d7708db56db1481%7C8370cf1416f34c16b83c724071654356%7C0%7C0%7C638199271199625943%7CUnknown%7CTWFpbGZsb3d8eyJWljoimC4wLjAwMDAiLCJQIjoiV2luMzliLCJBTiI6IjEhaWwiLCJXVCi6Mn0%3D%7C3000%7C%7C%7C&sdata=0l7RJVBVq%2B70v%2FN1XWTVDI3xIW5XwvGMBJgBEOr%2FyGQ%3D&reserved=0>) didn't show a "clinically meaningful" superiority of Machine Learning over Logistic Regression based statistics for stroke risk prediction, ML is indeed becoming a more available and useful tool, so the matter is valuable.

The statistical analyses of the study were well performed.

The major limitations are data availability and data ascertainment, as well as in any other study with this kind of dataset.

My biggest concern is that I didn't perceive any clear differential of this study over other stroke ML prediction risk initiatives

(<https://eur03.safelinks.protection.outlook.com/?url=https%3A%2F%2Fpubmed.ncbi.nlm.nih.gov%2F36580703%2F&data=05%7C01%7Cvasa.curcin%40kcl.ac.uk%7Cec49d280ec734d3e1d7708db56db1481%7C8370cf1416f34c16b83c724071654356%7C0%7C0%7C638199271199625943%7CUnknown%7CTWFpbGZsb3d8eyJWljoimC4wLjAwMDAiLCJQIjoiV2luMzliLCJBTiI6IjEhaWwiLCJXVCi6Mn0%3D%7C3000%7C%7C%7C&sdata=f4zbyRbTHyygPKBLcHeEtguQnzBaHJY2aNdzAsHRKml%3D&reserved=0>), except for being cross-national (but still within the same continent - and similar ethnicity).

The comparison with other stroke ML prediction risk initiatives could be better explored in the discussion section.

Response: Thanks for the suggestion. We have updated the Discussion section with comparisons of the initiatives of other ML studies. The intended application of the model was to benchmark stroke services using mortality as an indicator of quality and safety. To achieve this, the models should be externally validated to make sure the models are generalisable. No other ML studies have been externally validated across different nations. We note that the generalisability in UK and Sweden might be due to similar population and care systems. It is not as certain whether this model would be generalizable to a nation with a much more different patient population or care delivery system. We have added these in Discussion.

VERSION 2 – REVIEW

REVIEWER	Huo, Zepeng
----------	-------------

	Stanford University, Biomedical Informatics
REVIEW RETURNED	05-Jul-2023

GENERAL COMMENTS	The authors addressed most of my comments. There are still some minor revisions needed, such as reference number formatting (e.g. in page 16 the references numbers are sometimes not in superscript) and the resolution of some of figures is relatively low. But overall this version should suffice for me to make a recommendation for accept.
--

REVIEWER	Reed, Grant Cleveland Clinic
REVIEW RETURNED	17-Jul-2023

GENERAL COMMENTS	The authors have done a good job addressing my comments and questions. I recommend close proofreading to polish the manuscript and assure avoidance of run-on sentences. However, the paper has been significantly improved. Congrats on a very nice study.
---

VERSION 2 – AUTHOR RESPONSE